# Caffeine and Cisplatin Effectively Targets the Metabolism of a Triple-Negative Breast Cancer Cell Line Assessed via Phasor-FLIM

**DOI:** 10.3390/ijms21072443

**Published:** 2020-04-01

**Authors:** Stephanie M. Pascua, Gabrielle E. McGahey, Ning Ma, Justin J. Wang, Michelle A. Digman

**Affiliations:** 1Department of Biomedical Engineering, University of California, Irvine, CA 92697, USA; smpascua@uci.edu (S.M.P.); mcgaheyg@uci.edu (G.E.M.); 2Department of Surgery, Stanford University School of Medicine, Stanford, CA 94305, USA; ningma01@stanford.edu; 3Department of Computer Science, Stanford, CA 94305, USA; Jjwang01@stanford.edu; 4Laboratory for Fluorescence Dynamics, University of California, Irvine, CA 92697, USA

**Keywords:** caffeine, cisplatin, phasor-FLIM, energy metabolism, breast cancer

## Abstract

Triple-negative tumor cells, a malignant subtype of breast cancer, lack a biologically targeted therapy. Given its DNA repair inhibiting properties, caffeine has been shown to enhance the effectiveness of specific tumor chemotherapies. In this work, we have investigated the effects of caffeine, cisplatin, and a combination of the two as potential treatments in energy metabolism for three cell lines, triple-negative breast cancer (MDA-MB-231), estrogen-receptor lacking breast cancer (MCF7) and breast epithelial cells (MCF10A) using a sensitive label-free approach, phasor-fluorescence lifetime imaging microscopy (phasor-FLIM). We found that solely using caffeine to treat MDA-MB-231 shifts their metabolism towards respiratory-chain phosphorylation with a lower ratio of free to bound NADH, and a similar trend is seen in MCF7. However, MDA-MB-231 cells shifted to a higher ratio of free to bound NADH when cisplatin was added. The combination of cisplatin and caffeine together reduced the survival rate for MDA-MD231 and shifted their energy metabolism to a higher fraction of bound NADH indicative of oxidative phosphorylation. The FLIM and viability results of MCF10A cells demonstrate that the treatments targeted cancer cells over the normal breast tissue. The identification of energy metabolism alteration could open up strategies of improving chemotherapy for malignant breast cancer.

## 1. Introduction

According to the National Breast Cancer Foundation, breast cancer is the most common cancer diagnosed in women, and one in eight women will be diagnosed with breast cancer in their lifetime [1]. A subtype of breast cancer is basal-like breast cancer, also known as triple-negative breast cancer (TNBC). Given its lack of estrogen receptors (ER), progesterone receptors (PR), and low expression of human epidermal growth factor receptor 2 (HER2), there is no effective biological targeted therapy [2]. MDA-MB-231 is a known representative of triple-negative breast cancer, which has aggressive behavior as they go through reattachment, cell metastasis, and cell aggregation [3]. There is a need for an effective therapy that treats triple-negative breast cancer [4,5].

Cisplatin is commonly used as a chemotherapy drug to treat different cancers today [6]. Cisplatin is a DNA cross-linking agent which induces apoptosis by introducing DNA damage through the distortion of the structure of the DNA duplex by binding covalently to the N^7^ position of purines to form 1,2- or 1,3-intrastrand crosslinks and interstrand crosslinks [7]. During cisplatin’s DNA damage mechanism, chlorines in the platinum compound allow the platinum to attach to guanine’s N^7^ position, cross-linking DNA strands. At a high enough concentration of cisplatin, the cells cannot repair the damage cisplatin has done and undergo apoptosis [7]. Neoadjuvant cisplatin has been proven to be efficacious in treating testicular, ovarian, and bladder cancers [6,8,9]. However, cisplatin also damages non-cancerous cells [10]. Therefore, it is important to find a treatment that specifically targets cancer cells. Despite the short-term results, the effectiveness of cisplatin declines as the cancer cells becomes more resistant to the drug [11].

Caffeine is a common chemical found in our daily diets, which is both a central nervous stimulant and a protein kinase inhibitor [12,13]. Caffeine affects specific protein kinases, including ATM and ATR, which play key roles in DNA damage repair that induce cell cycle arrest at the G1 phase and apoptosis signaling [14]. Studies have shown that the combination of caffeine and cisplatin is an even more effective treatment than cisplatin alone [11,15,16]. Caffeine has received considerable attention in the past decade because of its ability to inhibit carcinogenesis in the lungs, skin, and ovaries [17,18,19,20]. The aim of this paper is to characterize the enhancing effects of caffeine effects in altering energy metabolism specifically to triple-negative breast cancer cells, which are characterized by a high frequency of mutations in ATM and BRCA2, as well as RB1 loss and cyclin E1 amplification [21,22]. These mutations also alter the function of mitochondria and energy metabolism [23,24].

Cancer cells undergo the Warburg effect, a shift in metabolic behavior away from mitochondrial oxidative phosphorylation (OXPHOS), towards aerobic glycolysis (GLY), even in the presence of excess oxygen [25]. Therefore, it has been of great interest to target the mitochondrial function as well as energy-related metabolic pathways for therapeutic development [26]. Cisplatin, a well-known platinum-based chemotherapeutic drug, is widely used as one of the major therapeutic options against cancer due to its ability to activate the DNA damage response and the induction of mitochondrial apoptosis exerts. After the initial therapeutic success associated with partial responses or disease stabilization, cisplatin treatment often results in the development of chemoresistance, leading to therapeutic failure. Approaches that can reverse cancer cell resistance to cisplatin treatment need to be explored. This study utilizes a phasor fluorescence lifetime imaging microscopy (phasor-FLIM) approach to evaluate caffeine and cisplatin’s ability to degrade a triple-negative cell line’s metabolic profile in comparison to a normally proliferating, or wild-type cell. Previous studies revealed that the presence of caffeine caused a decrease of cisplatin-induced cell cycle arrests at the S and the G2 phase in lung cancer [11]. FLIM provides a state-of-art way to measure the decay curve of a fluorescent species without fitting and identify biochemical reactions like oxidation and reduction, which can be used to map the effects of the potential drug in real-time without perturbing the cell behavior [6]. The coenzyme nicotinamide adenine dinucleotide (NADH) is the principal electron acceptor in glycolysis and electron donor in oxidative phosphorylation, which has been found to have endogenous fluorescence within the cells [27]. The protein-bound form of NADH is associated with energy production through oxidative phosphorylation with a lifetime of 3.4 ns (bound with lactate dehydrogenase) and maps on the left top of the phasor plot, whereas the free form of NADH is linked to glycolysis with a lifetime of 0.38 ns which maps on the right bottom of the phasor plot [28].

This change in signature correlates to a shift towards OXPHOS and GLY, respectively, which has been previously described as the metabolic or M-trajectory [27,28,29,30]. This trajectory has also been shown to correlate with results found in conventional biochemical assays when OXPHOS or GLY inhibitors are used to shift metabolic signatures towards one another [27,31,32,33]. To ensure the collection of the fluorescence lifetime of NADH, all of the measurements were done using a 2-photon laser excitation set to 740 nm, and the emission was collected with an emission bandpass filter (460/40 nm) to select the emission of NADH. Our study focuses on the energy metabolism of three cell lines: triple-negative breast cancer (MDA-MB-231), estrogen-receptor lacking breast cancer (MCF7), and normal breast epithelial cells (MCF10A) with caffeine, cisplatin, or a combination of the two. We have found that caffeine alters energy metabolism and significantly increases the effectiveness of the treatment of cisplatin for MDA-MB-231 and MCF7. On the other hand, MCF10A cells were not affected by any of the three treatments. These results indicate that monitoring the effects of caffeine together with DNA damaging agents on energy metabolism in different breast cancer cells can provide effective therapy, which raises the possibility of pharmacologic targeting of triple-negative breast cancers.

## 2. Results

### 2.1. Dosage Dependency of Caffeine

Cancer cells are well equipped to regulate and maintain energy metabolism. In particular, proliferative cells require a small increase of ATP compared to a larger need for precursor molecules and NADPH for maintaining biogenesis [34]. Thus, we aimed to use the phasor-FLIM method to quantify the bound NADH fraction as an indicator of cellular response. Given that triple-negative breast cancer cells are considered to be more aggressive and have a poorer prognosis than other types of breast cancer within our experiment, we treated MDA MB-231 cells to determine the optimal baseline for metabolic shifts on malignant cancer cells. The cells were treated with caffeine at the concentrations of 0 nM, 2 nM, 8 nM, 32 nM, 125 nM, 500 nM, 2000 nM, 8000 nM, for an hour (Figure 1). From concentrations 0 to 32 nM, the caffeine does not induce a significant shift in energy metabolism or the free NADH fraction until it reaches 125 nM, where it starts to produce a response of induced death in the MDA-MB-231 cell line. There is a significant drop of bound NADH fraction at 125 nM of caffeine, which indicates that anything higher than this concentration introduces a significant glycolytic response from the cells. Concentrations at 2000 nM or higher can lead to cell death. Lethality was determined by studying cell morphology after one hour of treatment. Healthy MDA-MB-231 cells’ morphology is elongated and oval, while the dead cells are detached from the substrate with a round shape [35]. Furthermore, 125 nM was chosen as the optimal concentration to stimulate the alteration of energy metabolism without killing the cells.

The graph presents a trend of the measured bound NADH fraction (*y*-axis) found within the MDA-MB-231 cells depending on the concentration of caffeine (*x*-axis) given. Each diamond represents the average bound NADH fraction at the specific concentration. The error bars correspond to the standard deviation of each group. The Student t-test *p*-values of the bound NADH fraction compare the treated group with the control group (0 nM): 2 nM = 0.493, 8 nM = 0.217, 32 nM = 0.063, 125 nM = 0.0038, 500 nM = 0.00477, 2000 nM = 0.0029, 8000 nM = 0.0022. The star (*) indicates significant difference (*p*-value <0.0.5). Concentrations around 2 nM to 32 nM are ineffective, 125 nM to 2000 nM introduces an energy metabolism shift. It is worth noting that a value of 2000 nM or higher leads to cell death based on the morphology. Each group was treated with the stated concentration of caffeine for one hour.

### 2.2. Cisplatin Treatment

Cisplatin is a commonly used chemotherapeutic method to treat breast cancer. Studies have found that the optimal dosage of cisplatin is 20 μM with a four-hour treatment [36]. To verify the efficacy of a cisplatin-based chemotherapeutic approach with breast cancer, we treated the triple-negative breast cancer cell line (MDA-MB-231), estrogen-receptor lacking breast cancer line (MCF7) and normal breast epithelial cell line (MCF10A) with 20 μM cisplatin for four hours before the phasor-FLIM acquisition (Figure 2). We collected the fluorescence intensity images at 740 nm excitation and pseudo-colored the long lifetime component as cyan and the short lifetime component as red.

With the cisplatin treatment, MDA-MB-231 and MCF7 cells demonstrate a clear alteration of energy metabolism. MDA-MB-231 cells show a significant change of energy metabolism towards glycolysis with the rightward shift on the phasor plot indicating a higher fraction of free NADH (red). This shift was determined to be due to excessive glycolytic activity coming from the nucleus via a separate analysis of the whole cell, nucleus, and cytoplasm. MCF7 cells indicate the significant alteration of energy metabolism to OXPHOS with the leftward shift on the phasor plot indicating a lower fraction of free NADH (cyan). The MCF10A cell line showed no significant differences between the treated group and the control group. This might imply that regular breast cells do not multiply fast enough for them to be significantly affected by cisplatin.

### 2.3. Caffeine Treatment

To determine the efficacy of a caffeine based chemotherapeutic strategy, we treated the triple-negative breast cancer cell line (MDA-MB-231), estrogen-receptor lacking breast cancer line (MCF7) and normal breast epithelial cell line (MCF10A) with a 125nM concentration of caffeine for one hour. This treatment was determined in the previous experiment to be the optimal dose to change the energy metabolism of triple negative breast cancer (Figure 1). Then, we performed phasor-FLIM acquisition (Figure 3, Appendix A).

The MDA-MB-231 and the MCF 7 cell line experienced a significant leftward shift in each of their respective phasor plots after the caffeine treatment. This shift demonstrates a more OXPHOS metabolism after the caffeine treatment is applied in both cell lines. This alteration of energy metabolism also appears in the FLIM images of each cell line as, after the caffeine treatment, both the MDA-MB-231 and the MCF7 cells show more cyan signal. This increase in cyan coloring is indicative of a higher fraction of free NADH and a longer lifetime compared to their respective controls. The MCF10A has no significant shift with 125 μM caffeine treatment.

### 2.4. Cisplatin and Caffeine Treatment

The combination treatment experiment was conducted to determine whether caffeine was able to enhance the effect of the cisplatin. We treated triple-negative breast cancer cell line (MDA-MB-231), estrogen-receptor lacking breast cancer line (MCF7) and normal breast epithelial cell line (MCF10A) with 20 μM cisplatin for four hours. For the last hour of the cisplatin treatment, caffeine was added to create a 125nM concentration treatment. Then we performed phasor-FLIM acquisition (Figure 4, Appendix A).

The MDA-MB-231 breast cancer cell line demonstrates a more OXPHOS metabolism after the cisplatin and caffeine treatment is applied. This is also indicated in its phasor plot where there is a significant leftward shift after applying the treatment. This leftward shift is indicative of a higher fraction of free NADH and a longer lifetime compared to its control (Figure 4a, more cyan). The MCF7 and MCF10A do not appear to have a significant trend with the double treatment (Figure 4c–f, and Appendix A).

### 2.5. Cell Viability with Different Treatments

To test whether caffeine, cisplatin or the combination of both induced cell death with the triple-negative breast cancer cell line (MDA-MB-231), estrogen-receptor lacking breast cancer line (MCF7) and normal breast epithelial cell line (MCF10A), cell viability was examined by trypan blue exclusion. There is a statistical significance between the control (untreated) and caffeine (125nM, one hour) and cisplatin (20 μM, four hours) combination treatment for the MDA-MB-231 cells, which decreased in the cell survival rate (Figure 5). None of the treatments caused significant cell death within MCF7 cells or MCF10A cells. This demonstrates that the combination treatment with caffeine enhances the effectiveness of cisplatin for targeting triple negative breast cancer cells without perturbing normal epithelial breast cells.

The graph shows the number of cells that survived the treatment. The MDA-MB-231 show the non-treated control, cisplatin, caffeine, and cisplatin plus caffeine results with blue, red, green, and purple respectively. The *p*-values were taken with respect to the control (*p*-value: Cisplatin group, 0.064, Caffeine group, 0.031, Cisplatin and Caffeine group, 0.1412). Star (*) indicates a significant difference with *p*-value <0.05. The MCF7 show the control, cisplatin, caffeine, and cisplatin plus caffeine results with blue, red, green, and purple respectively. The *p*-values were taken with respect to the control (*p*-value: Cisplatin group, 0.3725, Caffeine group, 0.8553, Cisplatin Caffeine group, 0.7877). The MCF10A show the control, cisplatin, caffeine, and cisplatin plus caffeine results with blue, red, green, and purple respectively. The *p*-values were taken with respect to the control (*p*-value: Cisplatin group, 0.4030, Caffeine group, 0.6895, Cisplatin Caffeine group, 0.2416). This experiment has been scientifically repeated four times.

## 3. Discussion

Triple negative breast cancer is highly aggressive due to the absence of receptors. Chemotherapy remains the mainstay for the treatment of triple negative breast cancer. Since cancer cells utilize glycolysis as the major metabolic process for energy production, targeting the energy-related metabolic pathways has renewed interest in the development of anti-cancerous approach [37]. Cisplatin is an anticancer drug that has been used to treat many cancer types including triple negative breast cancer, which can activate DNA damage response and induce energy metabolism alteration and mitochondrial apoptosis. However, cisplatin-treated cells usually develop drug resistance. To improve treatment efficiency, we introduce caffeine coupled with cisplatin-based cancer treatment to explore the effect on cancer prevention and treatment.

Here, we report that caffeine has effectively enhanced the cisplatin chemotherapeutic treatment activity for the triple negative breast cancer cell MDA-MB-231. First, we found MDA-MB-231 cells exhibit a dose-dependent metabolic and morphological reaction to caffeine (Figure 1). The 125 nM concentration was determined to be the optimal concentration. With the treatment of 32nM or less of caffeine, the MDA-MB-231 cells did not experience a significant shift in energy metabolism or any morphological changes that would be consistent with cell death, rounding or blebbing. [35]. With a treatment of 125 nM caffeine, the cells start to show a significant response to the treatment, seen in the energy metabolism shift towards a higher fraction of bound NADH indicates a higher level of oxidative phosphorylation. These changes by caffeine could be due to caffeine’s activity as a stimulant triggering the release of more energy [12]. With treatments larger than 2000 nM caffeine, the cells experience a change in their morphology from elongated to round. Apoptosis might be triggered because the caffeine concentrations are too high for the cells to recover, as it stays in G1 arrest leading it to cell death [16]. Second, the 125 nM concentration of caffeine clearly caused an effect in the MDA-MB-231 and the MCF7 energy metabolism profile but not the MCF10A (Figure 3). This indicates that the caffeine may be able to alter the energy metabolism in cancerous cells without harming the surrounding tissues. Third, the cisplatin treatment shifted the MDA-MB-231 cells towards glycolysis, which was initially unexpected. However, it is possible that this glycolytic shift is due to the nucleus repairing itself. Since MDA-MB-231 cells replicate more quickly, they might be experiencing DNA damage while they have active DNA repair mechanisms running, thus withstanding the cisplatin treatment and localizing glycolytic activity around the nucleus to sustain this DNA repair. However, the MCF7 and the MCF10A cells were minimally affected. Lastly, the combination treatment of caffeine and cisplatin affects both MDA-MB-231 and MCF7 cells, shifting both of them towards OXPHOS. This shift in energy metabolism is indicative of cell death in both of these cell lines. This means that the treatment kills the cancerous cells. Additionally, the phasor plot and the cell viability tests show that the non-cancerous cells were not significantly affected. We used fluorescence lifetime imaging microscopy to measure all these metabolic alterations. This method could help to monitor the real-time response after treatment non-invasively and longitudinally. To the best of our knowledge, this is the first time FLIM has been used for drug screening.

In conclusion, we have found that caffeine combined with cisplatin creates an effective chemotherapeutic treatment for breast cancer by shifting the energy metabolism of MDA-MB-231 and MCF7, two known breast cancer lines, and significantly increases the efficiency of cisplatin to treat the cancerous cells. This provides insight to develop therapies with caffeine consumption to bring many beneficial effects on breast cancer treatment. Phasor-FLIM also provides a real-time and quantitative measurement of redox metabolic phenotypes and potentially applies to long-term drug response prediction.

## 4. Materials and Methods

### 4.1. Fluorescence Lifetime Imaging Microscopy (FLIM)

Fluorescence lifetime images of the caffeine dose dependency experiment were acquired on Zeiss LSM880 (Carl Zeiss, Jena, Germany), a multi-photon microscope coupled with a Ti: Sapphire laser (Spectra-Physics Mai Tai, Mountain View, CA, USA) with 80 MHz repetition rate. The FLIM data detection was performed by the photomultiplier tube (H7422p-40, Hamamatsu, Japan) and an a320 FastFLIM FLIMbox (ISS, Champaign, IL, USA). The rest experiments were performed with an Olympus Fluoview. The cells were excited at 740 nm; an average power of ~5 mW was used as previously in live cells and tissue [38]. An Olympus UplanSApo 60×/1.2 NA water objective was used. The following settings were used for the FLIM data collection: image size of 256 × 256 pixels, a scan speed of 20 µs/pixel. A dichroic filter at 690 nm was used to separate the fluorescence signal from the laser light. Additionally, the emission signal was split with a 496 nm LP filter and detected in two channels using a bandpass filter 460/80 and a 540/50 filter. Every FLIM image was acquired for 50 frames of the same field of view with 256 × 256 per frame. Only the blue channel (460/80) data were used for this study. FLIM calibration of the system was performed by measuring the known lifetime of a fluorophore coumarin 6 (dissolved in ethanol), which has a known fluorescence lifetime of τ = 2.5 ns [39]. Embryos were kept in standard culture conditions, 37 °C and at 5% CO2. FLIM data were acquired and processed by the SimFCS software developed at the Laboratory of Fluorescence Dynamics (LFD).

### 4.2. Phasor Approach

Phasor-FLIM is an approach to analyze the data by calculating the decay curve of a fluorescent species without the use of fitting.

In FLIM, the raw data (intensity at each pixel) are transformed into polar coordinates by plotting the sine and cosine using the Fourier transformation. The g and s coordinates represent the decay curve at each pixel of the image. The coordinates are given by: $$g_{i}(\omega )=\int_{0}^{\infty }I\left(t\right)cos\left(\omega t\right)dt/\int_{0}^{\infty }I\left(t\right)dt$$
$$s_{i}(\omega )=\int_{0}^{\infty }I\left(t\right)sin\left(\omega t\right)dt/\int_{0}^{\infty }I\left(t\right)dt$$

Therefore, a phasor analysis transforms complicated spectrum and decay into a simple and unique position on the phasor plot.

### 4.3. Cell Preparation

Three cell lines: malignant breast cancer cell lines MDA-MB-231, benign breast cancer cell lines MCF7 and normal breast epithelial cell lines MCF10A were used in the experiments. The cells were maintained and cultured in a 37 °C incubator. MDA-MB-231 and MCF7 cells were placed in cell culture flasks with DMEM media, 10% fetal bovine and 1% penicillin streptomycin. MCF10A cells were placed in DMEM/F12 media, 5% Hose Serum, 20 ng/mL EGF, 0.5 mg/mL Hydrocortisone, 100 ng/mL Cholera Toxin, 10 μg/mL Insulin and 1% 100x Pen/Strep. Twenty-four hours before the cells are imaged, the cells are trypsinized and placed into an eight-well chamber (cell viability assay) or 35 mm imaging dish (Matek, treatment assay) once they are about 70% confluent. There should be two wells of the three different cell lines; one set of the cells should have 125 nM caffeine, 20 μM cisplatin or the combination of the two (experimental), the other having equal amounts of media to be consistent with the experimental volume (control). The caffeine requires an hour treatment and the cisplatin requires a four-hour treatment.

## Figures and Tables

**Figure 1 ijms-21-02443-f001:**
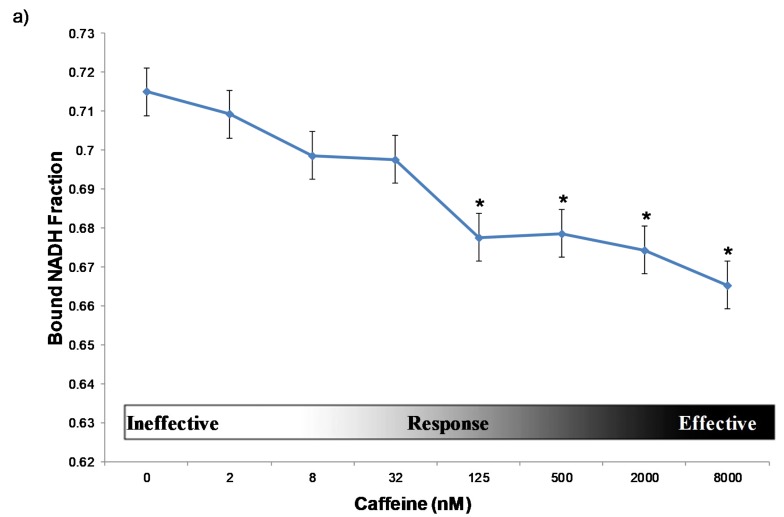
**The** efective dosage of caffeine on MDA-MB-231 cells. (**a**) Bound NADH fraction at varying concentrations of caffeine on MDA-MB-231 cells. The star (*) indicates significant difference (*p*-value <0.05) (**b**) Representative transmission (top) and FLIM image pseudo colored long lifetime as cyan and short lifetime as red (bottom) associated with each caffeine treatment (bottom) (scale bar: 20 μm).

**Figure 2 ijms-21-02443-f002:**
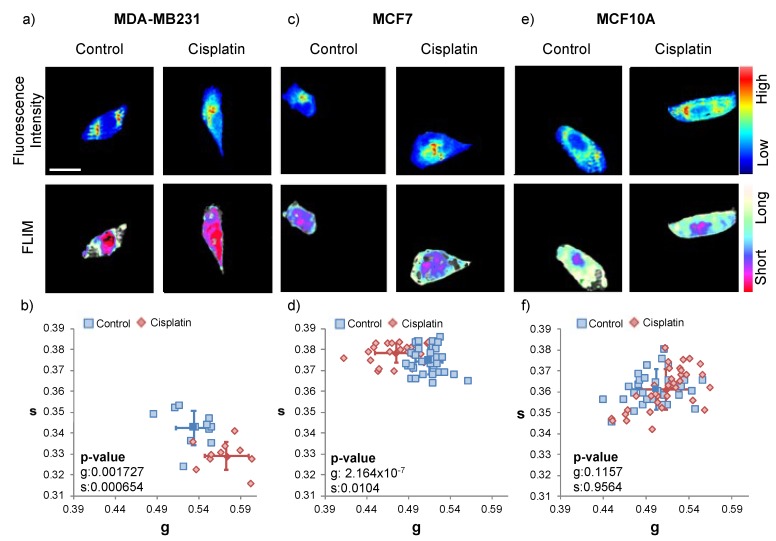
Fluorescence lifetime trajectories of the metabolic state of MDA-MB-231, MCF7, and MCF10A cells with 20 μM cisplatin treatment for four hours. (**a**) Fluorescence intensity image (top), and FLIM image pseudo colored long lifetime as cyan and short lifetime as red (bottom) for control and cisplatin-treated MDA-MB-231 cells; the scale bar is 20 μm, and all the images in this figure have the same scale. (**b**) The g and s values of control and cisplatin-treated MDA-MB-231. The blue squares are from the controls (untreated MDA-MB-231), red diamonds are from the cisplatin-treated MDA-MB-231, and the solid shapes (square and diamond) with the error bars correspond to the average and the standard deviation of each group (using the Student t-test to compare the treated group to the control group, *p*-values for g, s and free to bound NADH ratio are 0.001727, 0.000654, 0.000154, representatively). FLIM plot illustrates a rightward shift from long to short lifetimes. (**c**) Fluorescence intensity image (top), and FLIM image pseudo colored long lifetime as cyan and short lifetime as red (bottom) for control and cisplatin-treated MCF7 cells. (**d**) The g and s values of control and cisplatin-treated MCF7. The blue squares are from the controls (untreated MCF7), red diamonds are from the cisplatin-treated MCF7, and the solid shapes (square and diamond) with the error bars correspond to the average and the standard deviation of each group (using the Student t-test to compare the treated group to the control group, *p*-values for g, s and free to bound NADH ratio are 2.164 × 10^−7^, 0.0104, 2.34 × 10^−7^, respectively). The FLIM plot indicates a leftward shift from short to long lifetimes. (**e**) Fluorescence intensity image (top), and FLIM image pseudo colored long lifetime as cyan, and short lifetime as red (bottom) for control and cisplatin-treated MCF10A cells. (**f**) The g and s values of control and cisplatin-treated MCF10A. The blue squares are from the controls (untreated MCF10A), red diamonds are from the cisplatin-treated MCF10A, and the solid shapes (square and diamond) with the error bars correspond to the average and the standard deviation of each group (Student t-test *p*-value: g = 0.1157, s = 0.9564, free to bound NADH ratio = 0.0643). FLIM plot illustrates there is no significant shift. This experiment has been scientifically repeated three times. The two other repeats are in Appendix A.

**Figure 3 ijms-21-02443-f003:**
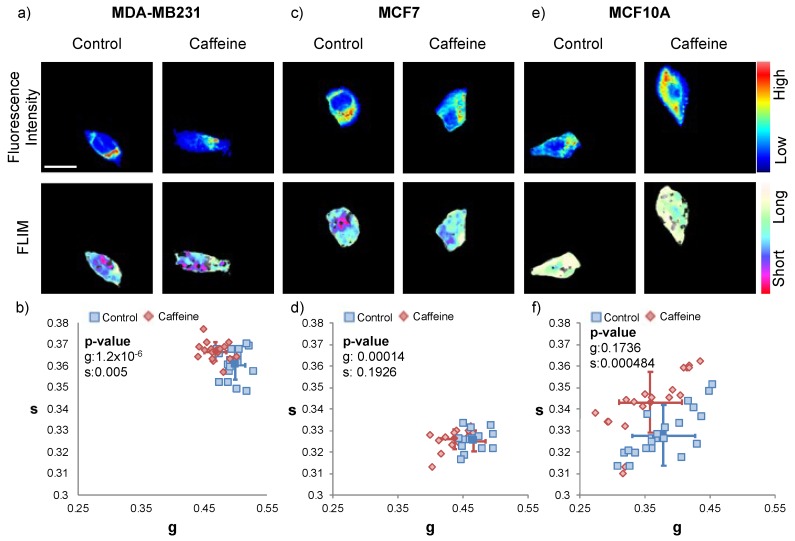
Fluorescence lifetime trajectories of the metabolic state of MDA-MB-231, MCF7, and MCF10A cells with 125 nM caffeine for one hour. (**a**) Fluorescence intensity image (top), and FLIM image pseudo colored long lifetime as cyan and short lifetime as red (bottom) for control and caffeine treated MDA-MB-231 cells; the scale bar is 20 μm, and all the images in this figure have the same scale. (**b**) The g and s values of control and caffeine treated MDA-MB-231. The blue squares are from the controls (untreated MDA-MB-231), red diamonds are from the caffeine treated MDA-MB-231, and the solid shapes (square and diamond) with the error bars correspond to the average and the standard deviation of each group (Student t-test *p*-values compare the treated group to the control group: g = 1.2 × 10^−6^, s = 0.005, free to bound NADH ratio = 9.52 × 10^−7^). FLIM plot illustrates a leftward shift from short to long lifetimes. (**c**) Fluorescence intensity image (top), and FLIM image pseudo colored long lifetime as cyan and short lifetime as red (bottom) for control and caffeine treated MCF7 cells. (**d**) The g and s values of control and caffeine treated MCF7. The blue squares are from the controls (untreated MCF7), red diamonds are from the caffeine treated MCF7, and the solid shapes (square and diamond) with the error bars correspond to the average and standard deviation of each group (using the Student t-test to compare the treated group to the control group, *p*-values for g, s and free to bound NADH ratio are 0.00014, 0.1926, 0.0031 respectively). FLIM plot indicates a leftward shift from short to long lifetimes. (**e**) Fluorescence intensity image (top), and FLIM image pseudo colored long lifetime as cyan and short lifetime as red (bottom) for control and caffeine treated MCF10A cells. (**f**) The g and s values of control and cisplatin-treated MCF10A. The blue squares are from the controls (untreated MCF10A), red diamonds are from the cisplatin-treated MCF10A, and the solid shapes (square and diamond) with the error bars correspond to the average and standard deviation of each group (using the Student t-test to compare the treated group to the control group, *p*-values for g, s and free to bound NADH ratio are 0.1736, 0.000484, 0.8028 respectively). FLIM plot illustrates there is no significant shift. This experiment has been scientifically repeated three times. The two other repeats are in Appendix A.

**Figure 4 ijms-21-02443-f004:**
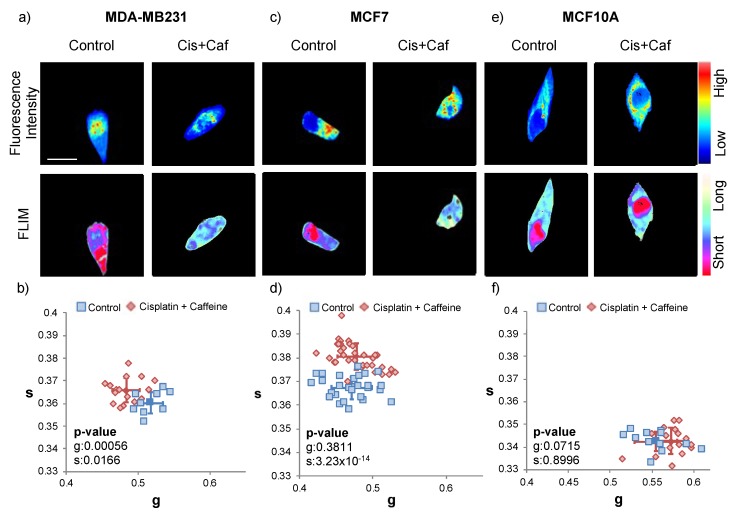
Fluorescence lifetime trajectories of the metabolic state of MDA-MB-231, MCF7 and MCF10A cells with caffeine and cisplatin treatment. (**a**) Fluorescence intensity image (top), and FLIM image pseudo colored long lifetime as cyan and short lifetime as red (bottom) for control and caffeine and cisplatin-treated MDA-MB-231 cells; the scale bar is 20 μm and all the images in this figure have the same scale. Note a shift in long to short lifetime correlates to blue to red in the FLIM images. (**b**) The g and s values of control and cisplatin and caffeine treated MDA-MB-231. The blue squares are from the controls (untreated MDA-MB-231), red diamonds are from the cisplatin and caffeine treated MDA-MB-231, and the solid shapes (square and diamond) with the error bars correspond to the average and standard deviation of each group (Student t-test *p*-value compare the treated group to the control group: g = 0.00056, s = 0.0166, free to bound NADH ratio = 3.63 × 10^−5^). FLIM plot illustrates a leftward shift from short to long lifetimes. (**c**) Fluorescence intensity image (top), and FLIM image pseudo colored long lifetime as cyan and short lifetime as red (bottom) for control and caffeine and cisplatin-treated MCF7 cells. (**d**) The g and s values of control and cisplatin and caffeine treated MCF7. The blue squares are the controls (untreated MCF7), red diamonds are the caffeine and cisplatin-treated MCF7, and the solid shapes (square and diamond) with the error bars correspond to the average standard deviation of each group (using the Student t-test to compare the treated group to the control group, *p*-values for g, s and free to bound NADH ratio are 0.3811, 3.23 × 10^−14^, 0.00187 respectively). FLIM plot indicates a rightward shift from long to short lifetimes. (**e**) Fluorescence intensity image (top), and FLIM image pseudo colored long lifetime as cyan and short lifetime as red (bottom) for control and caffeine and cisplatin-treated MCF10A cells. (**f**) The g and s values of control and cisplatin and caffeine treated MCF10A. The blue squares are the controls (untreated MCF10A), red diamonds are from the cisplatin and caffeine treated MCF10A, and the solid shapes (square and diamond) with the error bars correspond to the average and standard deviation of each group (using the Student t-test to compare the treated group to the control group, *p*-values for g, s and free to bound NADH ratio are 0.0715, 0.8996, 0.162 respectively). FLIM plot illustrates there is no significant shift in the MCF10A cell line. This experiment has been scientifically repeated three times. The two other repeats are in Appendix A.

**Figure 5 ijms-21-02443-f005:**
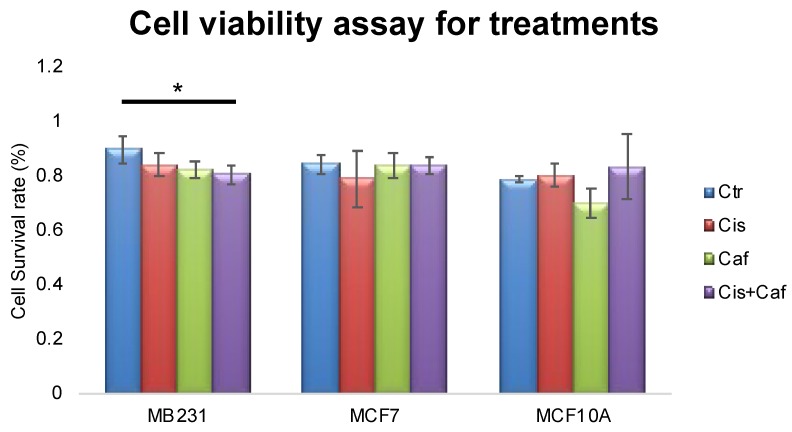
Cell viability assay demonstrates the cell survival rate for each treatment with MDA-MB-231, MCF7 and MCF10A cell lines. Star (*) indicates a significant difference with *p*-value <0.05.

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
