# Peer review of "Caffeine and Cisplatin Effectively Targets the Metabolism of a Triple-Negative Breast Cancer Cell Line Assessed via Phasor-FLIM"

_ijms, 2020, doi:10.3390/ijms21072443_

Round 1
Reviewer 1 Report
This paper clearly shows that the short term treatment of breast cancer cell lines such as triple‐negative breast cancer (MDA‐MB231), estrogen‐receptor lacking breast cancer (MCF7) and breast epithelial cells (MCF10A) with caffeine shifts cell metabolism towards respiratory‐chain phosphorylation by measuring ratio of free to bound NADH using a phasor‐ fluorescence lifetime imaging microscopy (phasor‐FLIM).
However, its biological output in the form of cell survival/viability show no significant difference in MCF7 and MCF10A cells.
Further, it is important to show caffeine dose response curve for all cell lines since each cell line is different. And then use right combination of and cisplatin dose and caffeine dose for that cell line to observe biological response.
Besides, it seems that the cell survival reported in this study after cisplatin, caffeine and cisplatin+caffeine treatment as compare with control is less than 10% which also raises the question of its true biological significance. It is not clear whether these changes are permanent or reversible. Long term treatment experiments need to be perform to address this questions.
In addition to free to bound NADH read out for metabolic changes other metabolic marker need to be use to confirm this observation.
Changes in cell morphology of MDA‐MB231 cells need to be supported with its images as described in the results section of 'Dosage Dependency of Caffeine'.
Author Response
Reviewer #1
This paper clearly shows that the short term treatment of breast cancer cell lines such as triple‐negative breast cancer (MDA‐MB231), estrogen‐receptor lacking breast cancer (MCF7) and breast epithelial cells (MCF10A) with caffeine shifts cell metabolism towards respiratory‐chain phosphorylation by measuring ratio of free to bound NADH using a phasor‐ fluorescence lifetime imaging microscopy (phasor‐FLIM).
However, its biological output in the form of cell survival/viability show no significant difference in MCF7 and MCF10A cells.
Further, it is important to show caffeine dose response curve for all cell lines since each cell line is different. And then use right combination of and cisplatin dose and caffeine dose for that cell line to observe biological response.
The main focus of our manuscript is to optimize the treatment for triple negative breast cancer as is considered to be more aggressive and have a poorer prognosis than other types of breast cancer, in our case, MDA-MB231. The cell survival/viability result shows significant change for MDA-MB231(treated versus untreated) but not for MCF7 or MCF10A. We suggest it is due to the effect of the combination of cisplatin and caffeine (ATR inhibitor) on ATR pathway activation was distinct in luminal, in our case MCF7 and basal-like breast cancer cells, MDA-MB2311. In MDA-MB-231 cells, the combination of cisplatin and ATR inhibitor will introduce higher cytotoxic effect compare to MCF-7 cells. The results are the same as we expected. To make it clear, we revised the results section as outlined below:
“There is a statistical significance between the control (untreated) and caffeine (125nM, 1 hour) and cisplatin (20mM, 4 hours) combination treatment for the MDA-MB231 cells which decreased in the cell survival rate (Figure 5). None of the treatments caused significant cell death within MCF7 cells or MCF10A cells. This demonstrates the combination treatment with caffeine enhances the effectiveness of cisplatin for targeting triple negative breast cancer cells without perturbing normal epithelial breast cells.”
For the same reason, we changed Figure 1 (“Effective dosage of caffeine on MDA MB-231”) label to read “Ineffective, Response and Effective” instead of “death” to show the selection for the concentration of caffeine for MDA MB-231 cells alone. Given that we are focusing on targeting these aggressive tumor cells, we used this as an index for treating the other cell lines. In the results section we have highlighted the selected of this concentration to clarify the reasons to why other cell types were not screened. This reads as follows:
“Cancer cells are well equipped to regulate and maintain energy metabolism. In particular, proliferative cells require a small increase of ATP compared to a larger need for precursor molecules and NADPH for maintaining biogenesis. Thus, we aimed to use the phasor-FLIM method to quantify the bound NADH fraction, as an indicator of cellular response. Given that triple-negative breast cancer cells are considered to be more aggressive and have a poorer prognosis than other types of breast cancer within our experiment, we treated MDA MB-231 cells for base line for metabolic shifts.”
Besides, it seems that the cell survival reported in this study after cisplatin, caffeine and cisplatin+caffeine treatment as compare with control is less than 10% which also raises the question of its true biological significance. It is not clear whether these changes are permanent or reversible. Long term treatment experiments need to be perform to address this questions.
Thank you very much for your suggestion. One of our goal is to assess the cellular response in real-time and contribute to personalized therapy. Our future experiments will include the longitudinal response and treatment experiments to confirm the prediction we can achieve from short-term phasor-FLIM results. Since these experiments requires a long-term and high number of repeats as well as machine learning modeling to obtain a solid prediction model. These experiments fall outside of the scope of this paper.
In addition to free to bound NADH read out for metabolic changes other metabolic marker need to be use to confirm this observation.
We appreciate the reviewer’s comment. Fluorescence lifetime imaging microscopy (FLIM) has been shown to be a powerful technique to measure metabolic indices of live cells2-9. By looking at the fluorescence lifetime of nicotinamide adenine dinucleotide (NADH), a metabolite involved in oxidative phosphorylation and glycolysis, we can determine the population of free and bound NADH due to their difference in lifetime decay. To ensure the collection of the fluorescence lifetime of NADH, all of the measurements were done using a 2-photon laser excitation set to 740nm and the emission was collected with emission bandpass filter (460/40 nm) to select the emission of NADH. This will allow us to quantify the “metabolic trajectory”, known as the “M trajectory”, of the cell at every pixel of our image and determine if the cell is undergoing OXPHOS or GLY. This trajectory has also been shown to correlate with results found in conventional biochemical assays when OXPHOS or GLY inhibitors are used to shift metabolic signatures towards one another6,10-12. Specific changes in the introduction session are as follows:
“This trajectory has also been shown to correlate with results found in conventional biochemical assays when OXPHOS or GLY inhibitors are used to shift metabolic signatures towards one another 6,10-12.To ensure the collection of the fluorescence lifetime of NADH, all of the measurements were done using a 2-photon laser excitation set to 740nm and the emission was collected with emission bandpass filter (460/40 nm) to select the emission of NADH.”
Changes in cell morphology of MDA‐MB231 cells need to be supported with its images as described in the results section of 'Dosage Dependency of Caffeine'.
Thank you very much for your suggestion, we added the cell morphology of MDA-MB231 cells in the results section of 'Dosage Dependency of Caffeine'.
1 Gomes, L. R. et al. ATR mediates cisplatin resistance in 3D-cultured breast cancer cells via translesion DNA synthesis modulation. Cell death & disease 10, 459 (2019).
2 Ma, N., Digman, M. A., Malacrida, L. & Gratton, E. Measurements of absolute concentrations of NADH in cells using the phasor FLIM method. Biomedical optics express 7, 2441-2452 (2016).
3 Provenzano, P. P., Eliceiri, K. W. & Keely, P. J. Multiphoton microscopy and fluorescence lifetime imaging microscopy (FLIM) to monitor metastasis and the tumor microenvironment. Clinical & experimental metastasis 26, 357-370 (2009).
4 Bird, D. K. et al. Metabolic mapping of MCF10A human breast cells via multiphoton fluorescence lifetime imaging of the coenzyme NADH. Cancer Res 65, 8766-8773, doi:10.1158/0008-5472.CAN-04-3922 (2005).
5 Datta, R., Alfonso-García, A., Cinco, R. & Gratton, E. Fluorescence lifetime imaging of endogenous biomarker of oxidative stress. Sci Rep 5, 9848, doi:10.1038/srep09848 (2015).
6 Cinco, R., Digman, M. A., Gratton, E. & Luderer, U. Spatial characterization of bioenergetics and metabolism of primordial to preovulatory follicles in whole ex vivo murine ovary. Biology of reproduction 95, 129, 121-112 (2016).
7 Stringari, C. et al. Metabolic trajectory of cellular differentiation in small intestine by Phasor Fluorescence Lifetime Microscopy of NADH. Scientific reports 2, 568 (2012).
8 Sameni, S., Syed, A., Marsh, J. L. & Digman, M. A. The phasor-FLIM fingerprints reveal shifts from OXPHOS to enhanced glycolysis in Huntington Disease. Scientific reports 6, 34755 (2016).
9 Mah, E. J., Lefebvre, A. E., McGahey, G. E., Yee, A. F. & Digman, M. A. Collagen density modulates triple-negative breast cancer cell metabolism through adhesion-mediated contractility. Scientific reports 8, 17094 (2018).
10 Ma, N. et al. Label-free assessment of pre-implantation embryo quality by the Fluorescence Lifetime Imaging Microscopy (FLIM)-phasor approach. Scientific reports 9, 1-13 (2019).
11 Stringari, C. et al. Metabolic trajectory of cellular differentiation in small intestine by Phasor Fluorescence Lifetime Microscopy of NADH. Sci Rep 2, 568, doi:10.1038/srep00568 (2012).
12 Stringari, C. et al. Phasor approach to fluorescence lifetime microscopy distinguishes different metabolic states of germ cells in a live tissue. Proc Natl Acad Sci U S A 108, 13582-13587, doi:10.1073/pnas.1108161108 (2011).
Reviewer #2
The paper reports an investigation aiming to show that caffeine can effectively enhance the activity of cisplatin towards the triple negative breast cancer cells MDA‐MB231. First, it is shown that treatment with 125 nM caffeine causes significant energy metabolism shift of the cells towards a higher level of oxidative phosphorylation. In contrast, cisplatin treatment shifts the MDA‐MB231 cells towards glycolysis. Finally, the combination treatment of caffeine and cisplatin causes shift of MDA‐MB231 cells towards oxidative phosphorilation. In addition, cell viability tests show that caffeine can marginally increase the efficiency of cisplatin to treat the MDA‐MB231 cells. Fluorescence lifetime imaging microscopy (FLIM) was the only technique used to measure all these metabolic alterations and it is stressed that this is the first time that FLIM has been used for drug screening.
I am not competent in FLIM, however I noticed from inspection of Figures 2 - 4 that some time changes appear to be quite small. Very small appears to be also the effect of caffeine, cisplatin, or caffeine+cisplatin on viability of MDA‐MB231 cells (Figure 5).
Thank you very much for your advice. We have mentioned the p-value in all the figure legends and the left bottom corner of the scatter plots. Even though the difference seems moderate, but they are statistically significant different between groups.
In conclusion, given for granted the competence of the authors in conducting this type of experiments I found that the results are of limited significance. That the combination of caffeine and cisplatin can be more effective than cisplatin alone has already been reported (Ref.11, 15, and 16).
Thank you very much for your review. This study provides new insight for targeting specifically triple negative breast cancer cells which are difficult to target. In addition, we provide new sights into the effect of this treatment and metabolism. This study shows that there is a correlation with targeting highly aggressive and proliferative tumor cells by effectively shifting their energy metabolism when they are treated with caffeine and cisplatin so only, they become more vulnerable and responsive to the treatment.
To make this clear, we revised the discussion section as outlined below:
In results
“Cancer cells are well equipped to regulate and maintain energy metabolism. In particular, proliferative cells require a small increase of ATP compared to a larger need for precursor molecules and NADPH for maintaining biogenesis. Thus, we aimed to use the phasor-FLIM method to quantify the bound NADH fraction, as an indicator of cellular response. Given that triple-negative breast cancer cells are considered to be more aggressive and have a poorer prognosis than other types of breast cancer within our experiment, we treated MDA MB-231 cells for base line for metabolic shifts.”
Discussion section:
This provides insight to develop therapies with caffeine consumption to bring many beneficial effects on breast cancer treatment. Phasor-FLIM also provides a real-time and quantitative measurement of redox metabolic phenotypes and potentially apply to long-term drug response prediction.
Additional points:
Lines 44-45: The sentence “chlorines in the platinum compound attach to guanine’s N7 position, crosslinking DNA” can be misleading since it can mean that are chlorines, and not platinum, that attack to guanine's N7 positions.
Lines 64-66: The sentence “Cisplatin is a well‐known platinum‐based drug has been used as one of the major therapeutic options the activation of the DNA damage response and the induction of mitochondrial apoptosis exerts.” needs to be corrected.
Line 76: The expression “oxidation and reduction which can be used to measure map the effects of the potential drug” needs to be corrected.
Line 97: There is a mistake in the sequence “concentrations of 0 nM, 2 nM, 8 nM, 32 nM, 125 nM, 50 nM, 2000 nM, 8000 nM for an hour (Figure 1).”
Thank you very much for your suggestion. We have updated the manuscript with your advice.

Reviewer 2 Report
The paper reports an investigation aiming to show that caffeine can effectively enhance the activity of cisplatin towards the triple negative breast cancer cells MDA‐MB231. First, it is shown that treatment with 125 nM caffeine causes significant energy metabolism shift of the cells towards a higher level of oxidative phosphorylation. In contrast, cisplatin treatment shifts the MDA‐MB231 cells towards glycolysis. Finally, the combination treatment of caffeine and cisplatin causes shift of MDA‐MB231 cells towards oxidative phosphorilation. In addition, cell viability tests show that caffeine can marginally increase the efficiency of cisplatin to treat the MDA‐MB231 cells. Fluorescence lifetime imaging microscopy (FLIM) was the only technique used to measure all these metabolic alterations and it is stressed that this is the first time that FLIM has been used for drug screening. I am not competent in FLIM, however I noticed from inspection of Figures 2 - 4 that some time changes appear to be quite small. Very small appears to be also the effect of caffeine, cisplatin, or caffeine+cisplatin on viability of MDA‐MB231 cells (Figure 5).
In conclusion, given for granted the competence of the authors in conducting this type of experiments I found that the results are of limited significance. That the combination of caffeine and cisplatin can be more effective than cisplatin alone has already been reported (Ref.11, 15, and 16).
Additional points:
Lines 44-45: The sentence “chlorines in the platinum compound attach to guanine’s N7 position, crosslinking DNA” can be misleading since it can mean that are chlorines, and not platinum, that attack to guanine's N7 positions.
Lines 64-66: The sentence “Cisplatin is a well‐known platinum‐based drug has been used as one of the major therapeutic options the activation of the DNA damage response and the induction of mitochondrial apoptosis exerts.” needs to be corrected.
Line 76: The expression “oxidation and reduction which can be used to measure map the effects of the potential drug” needs to be corrected.
Line 97: There is a mistake in the sequence “concentrations of 0 nM, 2 nM, 8 nM, 32 nM, 125 nM, 50 nM, 2000 nM, 8000 nM for an hour (Figure 1).”
Author Response
Reviewer #2
The paper reports an investigation aiming to show that caffeine can effectively enhance the activity of cisplatin towards the triple negative breast cancer cells MDA‐MB231. First, it is shown that treatment with 125 nM caffeine causes significant energy metabolism shift of the cells towards a higher level of oxidative phosphorylation. In contrast, cisplatin treatment shifts the MDA‐MB231 cells towards glycolysis. Finally, the combination treatment of caffeine and cisplatin causes shift of MDA‐MB231 cells towards oxidative phosphorilation. In addition, cell viability tests show that caffeine can marginally increase the efficiency of cisplatin to treat the MDA‐MB231 cells. Fluorescence lifetime imaging microscopy (FLIM) was the only technique used to measure all these metabolic alterations and it is stressed that this is the first time that FLIM has been used for drug screening.
I am not competent in FLIM, however I noticed from inspection of Figures 2 - 4 that some time changes appear to be quite small. Very small appears to be also the effect of caffeine, cisplatin, or caffeine+cisplatin on viability of MDA‐MB231 cells (Figure 5).
Thank you very much for your advice. We have mentioned the p-value in all the figure legends and the left bottom corner of the scatter plots. Even though the difference seems moderate, but they are statistically significant different between groups.
In conclusion, given for granted the competence of the authors in conducting this type of experiments I found that the results are of limited significance. That the combination of caffeine and cisplatin can be more effective than cisplatin alone has already been reported (Ref.11, 15, and 16).
Thank you very much for your review. This study provides new insight for targeting specifically triple negative breast cancer cells which are difficult to target. In addition, we provide new sights into the effect of this treatment and metabolism. This study shows that there is a correlation with targeting highly aggressive and proliferative tumor cells by effectively shifting their energy metabolism when they are treated with caffeine and cisplatin so only, they become more vulnerable and responsive to the treatment.
To make this clear, we revised the discussion section as outlined below:
In results
“Cancer cells are well equipped to regulate and maintain energy metabolism. In particular, proliferative cells require a small increase of ATP compared to a larger need for precursor molecules and NADPH for maintaining biogenesis. Thus, we aimed to use the phasor-FLIM method to quantify the bound NADH fraction, as an indicator of cellular response. Given that triple-negative breast cancer cells are considered to be more aggressive and have a poorer prognosis than other types of breast cancer within our experiment, we treated MDA MB-231 cells for base line for metabolic shifts.”
Discussion section:
This provides insight to develop therapies with caffeine consumption to bring many beneficial effects on breast cancer treatment. Phasor-FLIM also provides a real-time and quantitative measurement of redox metabolic phenotypes and potentially apply to long-term drug response prediction.
Additional points:
Lines 44-45: The sentence “chlorines in the platinum compound attach to guanine’s N7 position, crosslinking DNA” can be misleading since it can mean that are chlorines, and not platinum, that attack to guanine's N7 positions.
Lines 64-66: The sentence “Cisplatin is a well‐known platinum‐based drug has been used as one of the major therapeutic options the activation of the DNA damage response and the induction of mitochondrial apoptosis exerts.” needs to be corrected.
Line 76: The expression “oxidation and reduction which can be used to measure map the effects of the potential drug” needs to be corrected.
Line 97: There is a mistake in the sequence “concentrations of 0 nM, 2 nM, 8 nM, 32 nM, 125 nM, 50 nM, 2000 nM, 8000 nM for an hour (Figure 1).”
Thank you very much for your suggestion. We have updated the manuscript with your advice.

Round 2
Reviewer 1 Report
Accept the paper in its current form.
Reviewer 2 Report
The authors have improved the layout of the manuscript by addressing most of the points raised by the Reviewer. Although the significance and the overall merit of the manuscript remain low, at this stage of the review process I don't think the paper can be greatly improved.